# Higher rank chirality and non-Hermitian skin effect in a topolectrical circuit

Penghao Zhu [1], Xiao-Qi Sun[1], Taylor L. Hughes[1]✉ & Gaurav Bahl [2]✉

While chirality imbalances are forbidden in conventional lattice systems, non-Hermiticity can effectively avoid the chiral-doubling theorem to facilitate 1D chiral dynamics. Indeed, such systems support unbalanced unidirectional flows that can lead to the localization of an extensive number of states at the boundary, known as the non-Hermitian skin effect (NHSE). Recently, a generalized (rank-2) chirality describing a 2D robust gapless mode with dispersion $\omega = k_x k_y$ has been introduced in crystalline systems. Here we demonstrate that rank-2 chirality imbalances can be established in a non-Hermitian (NH) lattice system leading to momentum-resolved chiral dynamics, and a rank-2 NHSE where there are both edge- and corner-localized skin modes. We then experimentally test this phenomenology in a 2-dimensional topolectric circuit that implements a NH Hamiltonian with a long-lived rank-2 chiral mode. Using impedance measurements, we confirm the rank-2 NHSE in this system, and its manifestation in the predicted skin modes and a highly unusual momentum-position locking response. Our investigation demonstrates a circuit-based path to exploring higher-rank chiral physics, with potential applications in systems where momentum resolution is necessary, e.g., in beamformers and non-reciprocal devices.

Chirality is a key characteristic of robust gapless modes—only by coupling a set of such modes with vanishing net chirality can we destabilize the modes and open an energy gap. For example, a 1D chiral mode is a one-way conducting channel with a linear dispersion $\omega = vk$. Its chirality is intrinsically defined as the sign of its group velocity $v$ without requiring any other constraints. To open a gap in a chiral channel one needs to backscatter and reverse the current flow, which can be accomplished only by coupling two such modes that have opposite group velocity/chirality. This robustness is heralded by the so-called chiral anomaly: subjecting a chiral channel to an electric field generates extra charges that break the conservation of the electric charge current by an amount proportional to the net chirality.

With crystalline symmetries, one can generalize the concept of chirality, robustness, and anomalies for gapless modes with more complex dispersion. Indeed, a generalized chirality for gapless modes with dispersion $\omega \sim k_x k_y$ has been recently proposed in mirror symmetric systems[1]. Such a chiral mode has a non-conserved momentum

current (momentum anomaly) in response to an electric field, and has a non-conserved charge current (charge anomaly) in response to a strain field. Since the charge anomaly is induced by a rank-2 (two-index) tensor gauge field (the strain tensor), this generalized chiral mode has been dubbed a rank-2 chiral mode. In contrast, the usual 1D chiral mode is anomalous in the presence of a rank-1 (vector) gauge field, and we therefore refer to it as a rank-1 chiral mode.

Although isolated chiral modes widely exist in the band structures of lattice systems, there is a no-go theorem excluding the lattice realization of a nonzero *net* chirality[2,3]. However, one can avoid the no-go theorem and observe unusual chiral dynamics on boundaries of topological phases[4], in periodically driven Floquet systems[5], or in non-Hermitian (NH) systems[6-8]. In the latter case, which is our focus, one can apply appropriate gain and loss to a lattice system that will effectively generate a chirality imbalance. Unidirectional flows are established in the long-time dynamics of such systems[9,10], and these lead to the localization of an extensive number of states at the boundary; a

[1]Department of Physics and Institute for Condensed Matter Theory, University of Illinois Urbana-Champaign, Urbana, IL 61801, USA. [2]Department of Mechanical Science and Engineering, University of Illinois Urbana-Champaign, Urbana, IL 61801, USA. ✉e-mail: hughest@illinois.edu; bahl@illinois.edu

phenomenon known as the non-Hermitian skin effect (NHSE)[11–15]. Heuristically, if there is a 1D chiral mode that is the most amplified (or least damped) in a NH system, then this mode is long-lived and can dominate the long-time dynamics. Such a mode will produce a net 1D chirality and hence a unidirectional particle current that accumulates density on a boundary thus forming the NHSE. More details about the relationship between the dynamics of long-lived modes and the NHSE can be found in Supplementary Material Section S1.

Generalizing to 2D systems, we construct a non-Hermitian lattice model hosting a rank-2 chiral mode in its long-time dynamics and illustrate the resulting unconventional momentum-resolved chiral dynamics and NHSE. Furthermore, we find skin modes on the edges and corners of our model on a square geometry. The localization of these skin modes has a mechanism different from previously reported higher-order skin modes[16] or hybrid skin-topological modes[17,18], and can be heuristically understood from the dynamics of the long-lived rank-2 chiral modes in the lattice model. We further implement our model in a topolectric circuit platform and experimentally probe key features of our model, including predictions of momentum resolved dynamics and a rank-2 NHSE.

## Results

### Rank-2 chiral mode

A rank-2 chiral fermion mode in 2D has the dispersion[1]:

$$E(\mathbf{k}) = \hbar v \xi k_x k_y - \mu, \tag{1}$$

where $v$ is a (Fermi) velocity, $\xi$ is a length scale, and $\mu$ is the Fermi energy. The isoenergy contours and dispersion relation for Eq. (1) are shown in Fig. 1. Importantly, if we impose a mirror symmetry about the line $x = y$ we can define a rank-2 chirality as $\chi_2 \equiv \text{sgn}(v\xi)$. Indeed, such a mirror symmetry guarantees that one cannot continuously deform

$k_x k_y \to -k_x k_y$ without breaking the symmetry, and hence $\chi_2$ is a fixed, well-defined sign.

It is illustrative to regard the rank-2 chiral mode as a collection of 1D chiral modes with a nontrivial chirality pattern. In particular we can describe the rank-2 chiral mode as a family of 1D chiral modes along the $x$-direction ($y$-direction) parameterized by $k_y$ ($k_x$), having a set of 1D chiralities given by $\chi_{1x}(k_y) = \chi_2 \text{sgn} \, k_y$ ($\chi_{1y}(k_x) = \chi_2 \text{sgn} \, k_x$). With this understanding, the robustness and the associated anomalies of a rank-2 chiral mode can be straightforwardly derived from the properties of the 1D chiral modes. As mentioned, a 1D chiral mode with a positive (negative) chirality has an anomalous charge conservation law proportional to the (opposite of the) external electric field[19]. For a uniform system the anomalous conservation law reduces to:

$$\partial_t \rho = \chi_1 \frac{e}{2\pi\hbar} E_x \tag{2}$$

where $\rho$ is the charge density. For example, if we turn on an electric field $E_x$ via the Faraday effect, e.g., by adiabatically shifting the $x$-component of the vector potential, $A_x$, by $h/eL$ where $L$ is the linear size of the system along $x$, then $\chi_1$ particles are generated. The rank-2 chiral mode is a collection of 1D chiral modes having momentum-dependent chiralities, though the net chiralities $\chi_X = \sum_{k_y} \chi_{1x}(k_y)$, $\chi_Y = \sum_{k_x} \chi_{1y}(k_x)$ in both directions vanish (as they must from time-reversal symmetry). As such a rank-2 chiral mode does not generate a net anomalous charge in an electric field. However, since the 1D chiral modes that comprise the rank-2 chiral mode with opposite chirality also have opposite transverse momenta, i.e., they have momentum-chirality locking, there will be an anomalous conservation law for the momentum density in the $i$-th direction $\rho_i$. Specifically, for a uniform system the anomalous conservation law

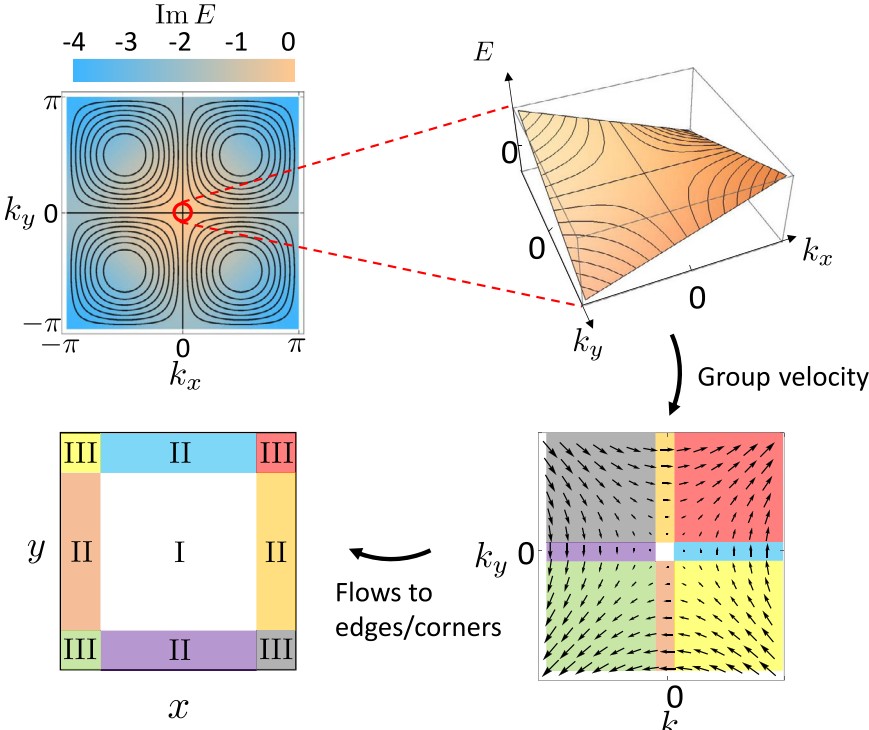

**Fig. 1 | Rank-2 chiral mode as a long-lived mode of a non-Hermitian lattice Hamiltonian and the resulting rank-2 NHSE.** The top left panel shows the iso-energy contours of $\text{Re} \, E = \sin k_x \sin k_y$ (black lines) and the loss $\text{Im} \, E = \cos k_x + \cos k_y - 2$. The top right panel manifests the dispersion of long-lived rank-2 chiral mode around $\Gamma = (0, 0)$. The group velocity field of the long-lived rank-2 chiral mode around $\Gamma = (0, 0)$ is shown in the bottom right panel. States with momenta in regions with different colors flow to different regions in spatial space, which leads to bulk-, edge-, and corner- localized modes as indicated in the bottom left panel.

reduces to:

$$\partial_t \begin{pmatrix} \rho_x \\ \rho_y \end{pmatrix} = \chi_2 \frac{e\Lambda^2}{4\pi^2} \begin{pmatrix} E_y \\ E_x \end{pmatrix}, \tag{3}$$

where $\Lambda$ is a momentum-cutoff scale[1].

We see from Eq. (3) that a rank-2 mode has an anomalous momentum response to a charge electric field. Remarkably, there is a related inverse effect where an anomalous charge response is produced by a "momentum" electric field. A momentum electric field is generated by a rank-2 tensor gauge field $\mathfrak{e}_\mu^a$ (for the translation symmetry) that couples to the momentum vector charge $k_a$ instead of the electric charge. Additionally, one can provide an interpretation of the translation gauge fields in terms of elasticity theory where $\mathfrak{e}_\mu^a = \delta_\mu^a - \frac{\partial u^a}{\partial x_\mu}$ where $u^a$ is the elastic displacement vector. Now, if we consider applying the momentum electric field $\mathcal{E}_x^y = \partial_x \mathfrak{e}_0^y - \partial_t \mathfrak{e}_x^y$ we find that modes at $k_y > 0 (k_y < 0)$ see an effective electric field in the $+\hat{x}(-\hat{x})$ direction, and analogously for $\mathcal{E}_y^x = \partial_x \mathfrak{e}_0^x - \partial_t \mathfrak{e}_y^x$. Thus, instead of the canceling effects of an ordinary electric field, in this case the contributions from opposite chiralities add up together. As shown in ref. [1] this leads to an anomalous conservation law (simplified for a uniform system):

$$\partial_t \rho = \chi_2 \frac{e\Lambda^2}{4\pi^2} (\mathcal{E}_x^y + \mathcal{E}_y^x). \tag{4}$$

To summarize, since for a rank-2 mode opposite momenta have opposite chiralities, but also see opposite effective electric fields in the presence of a momentum electric field $\mathcal{E}_x^y$ or $\mathcal{E}_y^x$, there is an anomalous contribution to the charge density.

## NH lattice model

To realize a non-zero rank-2 chirality in the long-time dynamics of a lattice system, we consider a single-band NH lattice model with Bloch Hamiltonian:

$$H(\mathbf{k}) = \sin k_x \sin k_y + i(\cos k_x + \cos k_y - m), \tag{5}$$

which necessarily has a mirror symmetry along $x = y$, and is reciprocal, i.e., $H(\mathbf{k}) = H^T(-\mathbf{k})$ where $H^T$ is the transpose of $H$. From the real and imaginary parts of the energy dispersion of Eq. (5), shown in the top left panel of Fig. 1, it is straightforward to see that the most long-lived mode (mode with largest imaginary energy) near the Fermi energy $E = 0$ is a rank-2 chiral mode with $\chi_2 = +1$ around the $\Gamma$-point in the Brillouin zone. Since the iso-energy (Fermi surface) contours of the rank-2 mode are open hyperbolas, our construction effectively lets us realize non-closed Fermi lines in a 2D lattice system. The group velocity of the long-lived rank-2 mode around the $\Gamma$ point is given by $\mathbf{v} = \partial \mathrm{Re}\, E(\mathbf{k})/\partial \mathbf{k} = (k_y, k_x)$, which suggests that a long-lived state with $(k_x, k_y)$ will contribute a current along the $(k_y, k_x)$ direction as illustrated in the bottom right panel of Fig. 1. Similar to the 1D case, since these currents are not compensated at long-times, they are expected to produce accumulated states localized on the edge and/or the corner as depicted in the bottom left panel of Fig. 1. To provide intuition for this expectation let us consider each case: (i) for $(k_x, k_y) = (0, 0)$ the corresponding state will be extended in the bulk (region I) because its group velocity is zero and thus it does not contribute to a unidirectional current, (ii) for states on the $k_x$ ($k_y$) axis the velocity is in the $y$-direction ($x$-direction), hence states will localize on the top/bottom (left/right) edges depending on the sign of the momentum (region II), and (iii) for states off the axes in one of the quadrants the states will localize near the four corners of the square (region III), because they have nonzero group velocity along both directions. Following this picture, in a system with $N^2$ sites, one expects to observe $O(1)$ bulk-localized modes, $O(N)$ edge-localized modes, and $O(N^2)$

corner-localized modes that correspond to states on points, lines, and areas in momentum space. More details about the topology related to this rank-2 NHSE can be found in the Supplementary Material Section S2. We emphasize that the rank-2 NHSE inherits the reciprocity of the Hamiltonian, i.e., states with opposite momentum localize on opposite boundaries. This is fundamentally different from the non-reciprocal NHSE associated with rank-1 chirality imbalances, where skin modes can only be found on one boundary. Furthermore, the observed corner skin modes are fundamentally different from previously studied higher-order skin modes[16] or hybrid skin-topological modes[17,18] and appear because of a new localization mechanism generated by the rank-2 chirality in our model. While a reciprocal NHSE has been previously observed in a 2D platform having exceptional points[20], higher-rank chiral modes and corner skin effects have not been previously reported in any system.

## Experimental implementation

We experimentally implemented the model in Eq. (5) using a topolectric circuit composed of passive elements as shown in Fig. 2—see also Methods. We use the theoretical foundation discussed in ref. [21] to map the real space hoppings in the reciprocal tight-binding model into a network of resistors, capacitors, and inductors (Fig. 2a, b). Additional details on the topolectric circuit design can be found along with a brief review in the Supplementary Materials Section S3. Using this approach, we constructed a circuit network containing $8 \times 8$ nodes (Fig. 2b, c), whose circuit Laplacian, $J$, implements the tight-binding Hamiltonian in Eq. (5), i.e., $J \propto -iH$. In this implementation, the dynamics generated by the Hamiltonian are mapped to a discrete process where we input a voltage and output a current in one step, and then convert the output current to a part of the input voltage for next step using an appropriate impedance normalization factor. Specifically, the input voltage in the $(n+1)$-th step (i.e., $V_{n+1}$) is generated by the input voltage in the $n$-th step (i.e., $V_n$) through $V_{n+1} = (1 + \alpha g_n J)V_n$, where $\alpha$ is the discrete "time" interval, and $g_n$ is the normalization factor. To illustrate an example for our circuit network in Fig. 2b, c, we choose $\alpha = 0.01$ and $g_n = 1/\max(JV_n)$ to simulate this discrete process, and we indeed observe the rank-2 dynamics discussed above, i.e., there are flows toward the edges and corners as shown in Fig. 2d.

For our measurements we first configure the topolectric circuit in a cylindrical geometry, that is, with a periodic boundary condition along $x$, and an open boundary along $y$ as shown in Fig. 3a. This is easily achieved with the help of appropriate wiring in the circuit. We measure the cross-impedance matrix $G_{ab} = V_a^{\mathrm{output}}/I_b^{\mathrm{input}}$ where $V_a^{\mathrm{output}}$ is the voltage at any node $a$ in response to input current $I_b^{\mathrm{input}}$ at any node $b$. This matrix $G$ is the inverse of the circuit Laplacian and therefore has the same eigenvectors. We can diagonalize $G$ to examine the eigenvectors of the circuit Laplacian. When we visualize the eigenstates of $G$, the NHSE is readily observed as an extensive number of eigenstates localized on the open boundaries. In the Supplementary Material Section S4 we discuss how the phase information encoded in the eigenstates can be used to confirm that this NHSE is momentum resolved.

To further observe the rank-2 phenomenology we configured the material with a square geometry where both the $x$ and $y$-directions are open. The measured $G$ matrix and its eigenstates are visualized in Fig. 3b. As expected from the intuitive picture (Fig. 1), we find the eigenstates localized in the bulk, edges, and corners of the system. A shortcoming of the intuitive picture in Fig. 1 is that it only considers states in a small energy/momentum window around $k_x = k_y = 0$, hence we cannot derive an exact counting rule for bulk-, edge-, and corner-localized modes for a finite open system. Even so, we observe that the order of the number of modes matches well with the expectations: there are $O(1)$ bulk-localized modes corresponding to states with group velocity $v_x = v_y = 0$, $O(N)$ edge-localized modes arising from states with group velocity $v_x \neq 0, v_y = 0$ and $v_x = 0, v_y \neq 0$, and $O(N^2)$

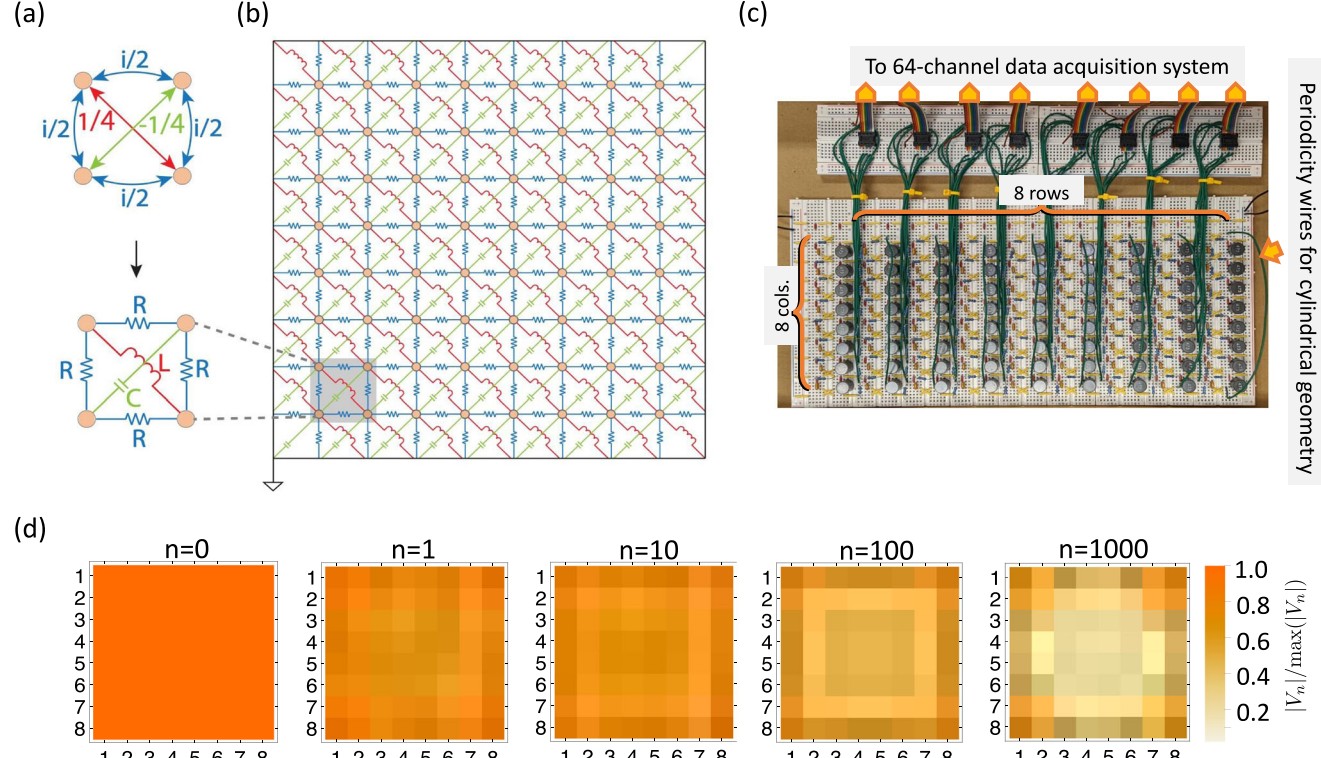

**Fig. 2 | Implementation of rank-2 chiral dispersion in a topolectric circuit. a** The real space hoppings of the tight-binding model $H(\mathbf{k}) = \sin k_x \sin k_y + i(\cos k_x + \cos k_y - m)$ [c.f. Eq. (5)] and their circuit implementations. Resistors $R$, inductors $L$, and capacitors $C$ are selected to satisfy the relationship $\omega C : \frac{1}{\omega L} : \frac{1}{R} = 1 : 1 : 2$ for a selected frequency (4.95 kHz). **b** Diagram for a 64-node circuit that is an implementation of the tight-binding model in Eq. (5) on a 8 × 8 square lattice. **c** Photograph of the assembled circuit board, of which each row/column corresponds to that in **b** directly. Each node is wired to a 64-channel data acquisition system. **d** Simulation showcasing the dynamical rank-2 behavior in the 64-node circuit network, where $n$ labels the step and $V_n$ (indicated by the color scale at each node) is the input voltage in $n$-th step.

corner-localized modes arising from states having group velocity $v_x, v_y \neq 0$ (where N=8 for our system).

Another key feature related to the rank-2 NHSE is that the momentum-chirality locking is converted to momentum-location locking (i.e., to edges and corners in the context of the rank-2 NHSE) in response to input excitations. To see this effect, we can experimentally apply voltage signals with relative phase differences that will introduce excitations with defined momenta (we note that the applied relative phase difference directly corresponds to the momentum in units of radians per node). In Fig. 4 we show the response for the cylindrical geometry configuration by exciting a row of the circuit with a nonzero $x$-directed momentum $k_x$. The measured output voltages explicitly show the momentum-location locking, i.e., as seen in Fig. 4, voltage inputs with positive (negative) momentum $k_x$ lead to a voltage accumulation on the top (bottom) edge. The relative phase measured between adjacent nodes along the respective edges matches the input signals, confirming the same $k_x$ momentum associated with the localized input. We also remark that the special case of $k_x = 0$ leads to only a symmetric voltage response around the excited row, i.e. it does not exhibit a preferred locking effect as is expected from our theory analysis. These observed responses to input voltages directly indicate a unidirectional momentum flow in an infinite system with nonzero rank-2 chirality, i.e., excitations with opposite momenta flow in opposite directions.

We next open both boundaries, as one would find in a, perhaps more practical, finite material (as configured in Fig. 2b). Even though the system is finite and momenta are not well defined, we can still consider the relative excitation phase as a proxy for $x$ and $y$ momenta $k_x$ or $k_y$. Results from various excitations are presented in Fig. 5. When applying excitations purely along $x$ or $y$ (Fig. 5a–c) we can confirm that

the momentum-location locking effect persists. As a curious aside, we surprisingly find that even though the voltage inputs with $\pm k_x$ or $\pm k_y$ on the boundary row or column lead to quite different configurations of output voltages (see Fig. 5b, c where we see some responses are extended into the bulk while some are extremely localized on the boundary), the effective load experienced by the excitation source is purely real and symmetric with respect to $k_x$ or $k_y$. Interested readers are referred to the Supplementary Material Section S5 for more details.

More interestingly, if we input signals having non-zero momentum simultaneously along both $x$ and $y$ directions we can see a voltage accumulation toward the corners as shown in Fig. 5d. This result once more verifies the intuitive picture about rank-2 NHSE. While we only present the amplitude responses here, the complete picture including the corresponding phase response can be found in Supplementary Material Section S6.

## Discussion

In this combined theoretical and experimental study, we have presented the 2D material that exhibits higher-rank chiral behavior. We specifically study both the rank-2 NHSE, as well as the remarkable momentum-resolved response that causes excitations in a finite system to lock to edges and corners. The responses observed in Fig. 4 and Fig. 5 directly confirm that the sign of group velocity of modes with momentum $(k_x, k_y)$ is given by $(\text{sgn}(k_y), \text{sgn}(k_x))$, and thus probes the rank-2 chirality. Since the rank-2 chirality determines the anomalous momentum and particle currents[1], the observed responses to input voltage signals can also be understood as an indirect probe of the charge and momentum anomalies due to a nonzero rank-2 chirality.

Looking forward, our circuit-based approach shows the potential for exploration of chiral physics in higher dimensions, including the

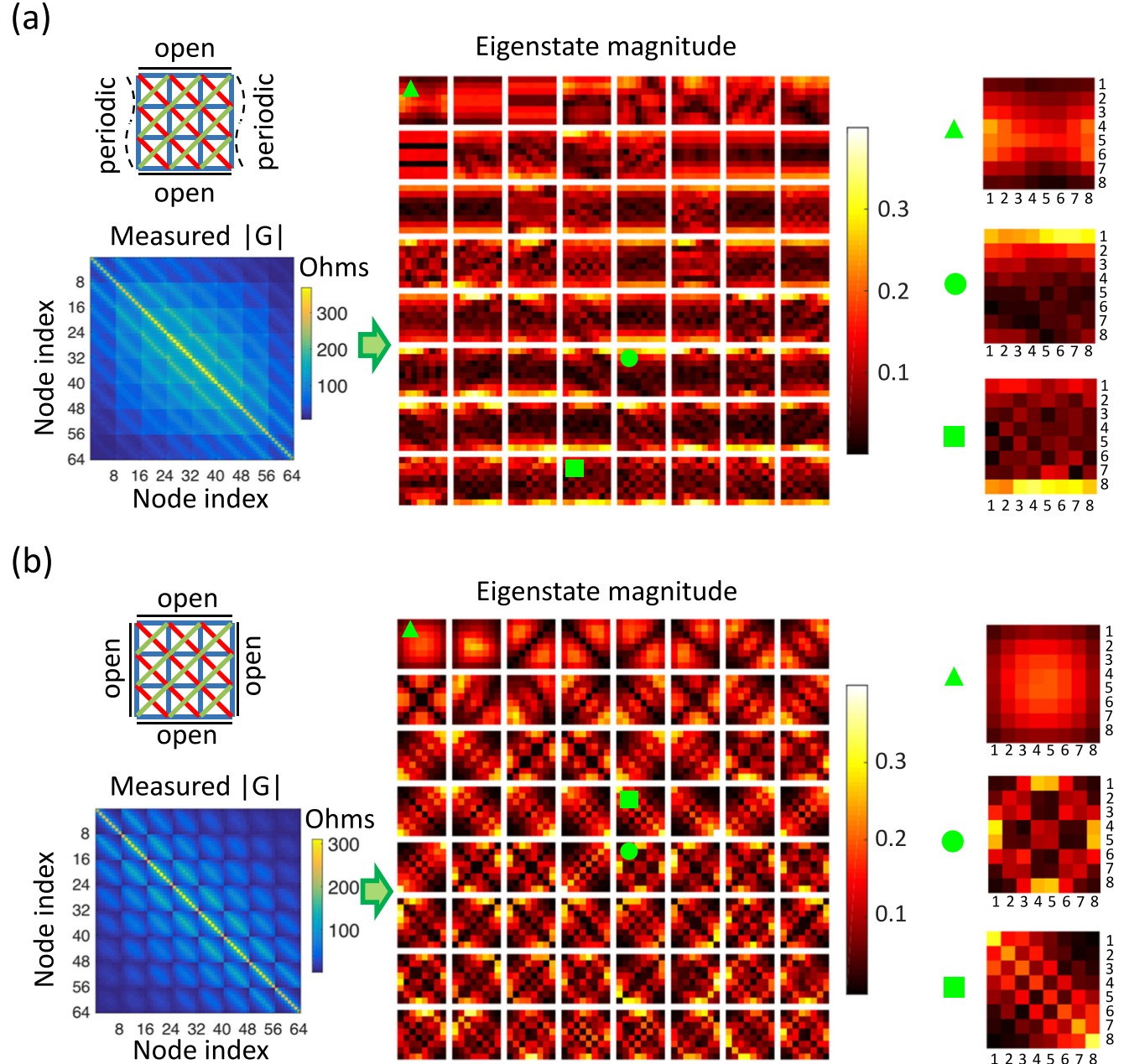

**Fig. 3 | Measurements of cross-impedance matrix *G* and its eigenstates under cylinder and square geometries. a** Visualization of the magnitude of measured matrix *G* (phase not presented) and its eigenstates under a cylinder geometry, where *x*-direction is periodic but *y*-direction is open as shown in the top left panel. The lines with different colors in the top left panels correspond to different hoppings previously shown in Fig. 2. Some representative bulk, top-edge, and bottom-edge localized eigenstates are zoomed in on the right. **b** The measured matrix *G* and its eigenstates under a square geometry, where both *x* and *y* directions are open as shown in the top left panel. Some representative bulk, edge, and corner corner modes are zoomed in on the right.

unusual responses to externally applied electromagnetic and geometric fields. Notably, this ability to resolve 2D vector momentum is particularly important in practical beamforming and sensor applications. Compared to the system with reciprocal NHSE discussed in ref. [20], our system can resolve the momentum along both directions in 2D since it is more isotropic. While the effects we show here are entirely reciprocal, the highly asymmetric momentum-resolved responses are also a key ingredient for producing non-reciprocal metamaterials.

## Methods

**Circuit construction:** The specific component values used for the topolectric circuit implementation were $L = 47$ mH ($\pm 5\%$), $C = 22$ nF ($\pm 1\%$), and $R = 732$ ohm ($\pm 1\%$). This allows the circuit Laplacian to

model the desired Hamiltonian at ~4.95 kHz. All components were assembled on solderless prototyping breadboards. **Data acquisition:** Experimental measurements were performed using Matlab and its built-in data acquisition toolbox. For cross-impedance matrix characterization, the drive currents were generated using a bench-top signal generator set to 4.95 kHz, sending its voltage output through a 497 ohm reference resistor. For the driven response tests, the phase-synchronized signals were generated using a National Instruments NI 9264 voltage output module. All response signals were captured using Keysight U2331A (64-channel) data acquisition hardware, with each channel set to one node of the 8x8 array. The magnitude and phase responses were measured against the drive signals using a digital lock-in that we coded in Matlab.

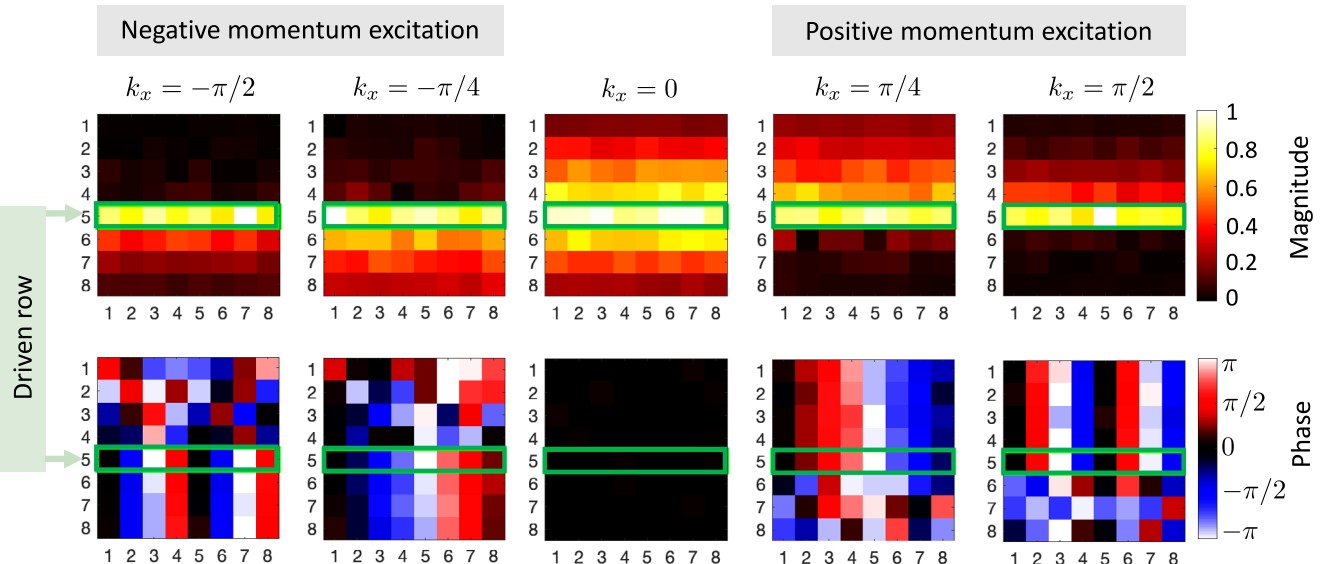

**Fig. 4 | Responses to input excitations for the cylinder geometry.** We apply voltage inputs on the fifth row (highlighted by the green box) with relative phase corresponding to momentum $k_x$. The specific momentum values are quantized by the periodicity and number of nodes in the structure. The top and bottom figures are paired and show the magnitudes and phase (relative to node driven with phase 0) of the output voltages for each case.

**Fig. 5 | Responses to input excitations for the square geometry.** The magnitude of voltage responses to different $k_x$ input on (**a**) the fifth row and (**b**) the first row are presented. **c** Similarly shows the magnitude of voltage responses to different $k_y$ input on the first column. **d** The magnitude of voltage responses to simultaneous $(k_x, k_y)$ input on the central square showing the corner-directed responses. Complete data sets including phase information are presented in the Supplementary Material Section S6.

## Data availability
The data that support the findings of this study are available from the corresponding author upon reasonable request.

## Code availability
The codes that support the findings of this study are available from the corresponding author upon reasonable request.

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

## Acknowledgements
This work was sponsored by the Multidisciplinary University Research Initiative (MURI) grant N00014-20-1-2325 and the US National Science Foundation EFRI grant EFMA-1641084. X.-Q.S. acknowledges support from the Gordon and Betty Moore Foundation's EPiQS Initiative through Grant GBMF8691. G.B. would additionally like to acknowledge support from the Office of Naval Research (ONR) Director for Research Early Career grant N00014-17-1-2209, and the Presidential Early Career Award for Scientists and Engineers. The authors thank Sasha Yamada for assistance with data acquisition.

## Author contributions
P.Z. and X.-Q.S. performed the theoretical study. P.Z. and G.B. designed the experiments. G.B. performed the experimental study. T.L.H. and G.B. supervised the work. All authors jointly wrote the paper.

## Competing interests
The authors declare no competing interests.
