## [Peer review file · Nature Communications]

REVIEWER COMMENTS

Reviewer #1 (Remarks to the Author):

In the manuscript, the authors address the topic of non-Hermitian skin effect in the context of electric circuit networks.

The first realization of skin effect in circuits was given in a 1D system in Helbig et al., Nature Physics 16, 747 (2020). Another realization, which is Ref. 18 in the present manuscript, was given in Hofmann et al., PRR 2, 023265 (2020).

As a new feature, the authors implement not just any kind of NH skin effect, but one that preserves reciprocity and that links back to higher-rank chiral states, a notation one of the co-authors had pioneered in the context of electrons in Weyl semimetals. Non-Hermiticity allows to circumvent the fermion doubling theorem and still a higher rank version of the skin effect allows for preserved reciprocity but a selective yet extensive localization of bulk modes at opposite terminations of the system.

In total, I believe this work to be interesting and innovative. In general, it is important that this manuscript communicates how easy it appears to be to implement highly involved condensed matter systems in circuits.

One central criticism before I could recommend publication and assess the degree of novelty: The system the authors implement and Ref. 18 appear very related: they both are in 2D, they show skin effect i.e. extensive mode localization, and they preserve reciprocity. It appears necessary to discuss in more detail the similarities and differences of Ref. 18 and this manuscript. In particular, in view of possible applications, it appears that both circuit realizations are extremely similar.

Reviewer #2 (Remarks to the Author):

The work reported in the manuscript "Higher rank chirality and non-Hermitian skin effect in a topoelectrical circuit" by Zhu et al. explores a system with rank-2 chirality imbalances. Such a system is not allowed in usual lattices due to the chiral-doubling theorem. Relying on this unique effect the

authors first propose and later experimentally verified the existence of both edge and corner-localized non-Hermitian skin (NHS) modes in a topoelectrical circuit.

The manuscript is written clearly and informative with a logical sequence, allowing the reader to follow the presented idea. The manuscript is original and unveiled many new effects, especially the momentum-position locking associated with different kinds of skin modes. This work advances the blooming field of non-Hermitian topological phases and deserves publication in Nature Communications. However, there are a few issues that needs to be addressed before it can be published.

1. It is not clear what is the role of the slowest decay in this work? All their results have nothing do with the decay rates of the modes. Neither they have measured it. Do they mean that in general the NHS modes have the slowest decay, which makes them most relevant? In any case, they should back it by some numerical results showing that in a finite system the skin modes have the slowest decay. Diagonalizing the circuit Laplacian should reveal it.

2. What is the underlying equation used to obtain Fig. 2(d)?

3. Related to the first point, do all modes show skin effect or only a few of them? This needs to be clarified at the beginning.

4. A projected band structure in a strip-geometry (PBC along x and OBC along y) will be helpful for understanding the results, especially for momentum resolved results in Figs. 4 and 5.

5. The NHS modes are usually associated with some topological invariants, such as the winding number. What is the topological invariant associated in this case?

Once the above points are addressed, I believe the manuscript can be accepted for publication.

NCOMMS-22-29445-T

Higher rank chirality and non-Hermitian skin effect in a topoelectrical circuit

Zhu, Sun, Hughes, and Bahl

November 20, 2022

We would like to thank the referees for providing valuable feedback on our work and finding our work “to be interesting and innovative” and (Ref. 1) “is written clearly and informative with a logical sequence” (Ref. 2).

Below we provide a detailed response to each Referee query and we believe that our manuscript has been significantly strengthened in the process.

1 Response to comments by the first Referee

R-1.1

In the manuscript, the authors address the topic of non-Hermitian skin effect in the context of electric circuit networks.

The first realization of skin effect in circuits was given in a 1D system in Helbig et al., Nature Physics 16, 747 (2020). Another realization, which is Ref. 18 in the present manuscript, was given in Hofmann et al., PRR 2, 023265 (2020).

As a new feature, the authors implement not just any kind of NH skin effect, but one that preserves reciprocity and that links back to higher-rank chiral states, a notation one of the co-authors had pioneered in the context of electrons in weyl semimetals. Non-Hermiticity allows to circumvent the fermion doubling theorem and still a higher rank version of the skin effect allows for preserved reciprocity but a selective yet extensive localization of bulk modes at opposite terminations of the system.

In total, I believe this work to be interesting and innovative. In general, it is important that this manuscript communicates how easy it appears to be to implement highly involved condensed matter systems in circuits.

A-1.1

We appreciate that the Referee finds our work interesting and innovative. In the following, we address the Referee’s main concern.

R-1.2

One central criticism before I could recommend publication and assess the degree of novelty: The system the authors implement and Ref. 18 appear very related: they both are in 2D, they show skin effect i.e. extensive mode localization, and they preserve reciprocity. It appears necessary to discuss in more detail the similarities and differences of Ref. 18 and this manuscript. In particular, in view of possible applications, it appears that both circuit realizations are extremely similar.

A-1.2

We appreciate this criticism and thank the referee for this suggestion. Our system is related to that in Ref. 18 because we both have momentum-dependent reciprocal NHSE. However, our work is quite different from Ref. 18 in the following aspects:

Figure 1: The spectra of our model with $m = 2$ under torus, cylinder, and square geometry.

- i As the Referee mentioned, our system is an implementation of the higher-rank chiral state which is a recently proposed state with novel dynamics and anomalous current and has not been discussed previously in any paper including Ref. 18.
- ii The model in Ref. 18 is a two-band model with exceptional points but our system has only one-band and has no exceptional points.
- iii Our system is more isotropic in the sense that we can have momentum-resolved skin effect along both directions. In contrast, the model in Ref. 18 can only have a k_y -resolved skin effect along the x -direction. Thus, our system enables a momentum sensor for 2D momentum (k_x and k_y) as an application, while the system in Ref. 18 can only be used as a momentum sensor for 1D momentum (k_y). Furthermore, in an open square geometry our system manifests skin modes localized near corners, which are absent in the system of Ref. 18. The corner skin modes in our system are also fundamentally different from previously discussed higher-order skin modes [Phys. Rev. B 102, 205118] or hybrid skin-topological modes [Phys. Rev. Lett. 123, 016805, Phys. Rev. B 102, 205118], and represent a new localization mechanism related to the rank-2 chirality in our model, of which the relation to topology bears future studies.
- iv Our system actually has a very unusual energy spectrum: in a torus geometry where both directions are periodic, we see in [Figure 1 (a)] a collection of loops (an example is highlighted by a blue oval in Figure 1 (a)) in the spectrum that manifest the momentum-dependent point gap winding. In a cylinder geometry where one direction is periodic and the other direction is open, most eigenstates have their energy on the imaginary axis as predicted by the generalized Brillouin zone theory [Figure 1 (b)]. However, there are some other states that have complex energies which form loops that cannot be understood from the canonical generalized Brillouin zone theory, which bears further investigations. Surprisingly, in the square geometry where both directions are open, the spectrum suddenly changes from a line on the imaginary axis to a region in the complex energy plane [Figure 1 (c)]. This phenomenon has not been observed in any other previously known systems (including the system in Ref. 18), and points to open questions that for future research.

To help emphasize the distinction between our work and previous work (especially Ref. 18) we added more sentences in lines 151-157. Also, we added a footnote in our last paragraph to emphasize that our system can resolve 2D momentum, but the system in Ref. 18 can only resolve momentum along one direction.

2 Response to comments by the second Referee

R-2.1

The work reported in the manuscript “Higher rank chirality and non-Hermitian skin effect in a topoelectrical circuit“ by Zhu et al. explores a system with rank-2 chirality imbalances. Such a system is not allowed in usual lattices due to the chiral-doubling theorem. Relying on this unique effect the authors first propose and later experimentally verified the existence of both edge and corner-localized non-Hermitian skin (NHS) modes in a topoelectrical circuit.

The manuscript is written clearly and informative with a logical sequence, allowing the reader to follow the presented idea. The manuscript is original and unveiled many new effects, especially the momentum-position locking associated with different kinds of skin modes. This work advances the blooming field of non-Hermitian topological phases and deserves publication in Nature Communications. However, there are a few issues that needs to be addressed before it can be published.

A-2.1 We thank the Referee for their positive comments about our manuscript, and are glad that they found our work to be original and well-written.

R-2.2

It is not clear what is the role of the slowest decay in this work? All their results have nothing do with the decay rates of the modes. Neither they have measured it. Do they mean that in general the NHS modes have the slowest decay, which makes them most relevant? In any case, they should back it by some numerical results showing that in a finite system the skin modes have the slowest decay. Diagonalizing the circuit Laplacian should reveal it.

A-2.2

We thank the Referee for this important physical question. In our manuscript we emphasized the idea of the slowest decaying modes as a theoretical motivation in analogy with the work in [Physical Review Letters **123** 206404 (2019)], i.e., intuitively the long-time dynamics will be dominated by the modes with the slowest decay. The dynamics of these modes can then be used to infer the location of skin modes as discussed in our third paragraph as well as in line 134-147 [also see our Ref. 8]. In another words, we use the dynamics of the slowest decay mode as an effective method to understand the NHSE and momentum-location locking in our model, and then in experiments we measure the proposed NHSE and momentum-location locking.

To make these points clearer, we have revised our fourth paragraph to clarify that we use the concept of the slowest decaying mode as a important guide to construct our model and understand its NHSE. We also added a section in the Supplementary material to explain the relationship between the dynamics of the slowest decaying modes and the NHSE.

R-2.3

What is the underlying equation used to obtain Fig. 2(d)?

A-2.3

The underlying equation is $V_{n+1} = (1 + \alpha g_n J)V_n$, where V_n is the input voltage in the n -th step, α is the discrete “time” interval, and g_n is the normalization factor. This equation is shown in line 171-172. The topo-electric formalism identifies the circuit Laplacian J with a Hamiltonian $-iH$. Using this we calculated a discrete evolution process generated by J on the circuit. More details for producing Fig. 2(d) can be found in line 173-176.

R-2.4

Related to the first point, do all modes show skin effect or only a few of them? This needs to be clarified at the beginning.

A-2.4

We mention in our third paragraph that there are an extensive number of eigenstates of the open boundary Hamiltonian that are localized at the boundary, which is the conventional manifestation of a NHSE (line

50-52). We can also determine a bit more. Indeed, for a system with $N \times N$ sites, our model the number of skin modes localized on the edges is of order $O(N)$, while the number of corner localized skin modes is of order $O(N^2)$ as discussed in line 134-145. Heuristically this skin mode counting is arrived at by considering the group velocities of the modes in a neighborhood of the rank-2 saddle point dispersion. In such a neighborhood, modes on the nodal lines $k_x = 0$ or $k_y = 0$ travel up/down or left/right and hence move to the y -normal and x -normal edges respectively. Modes that have both k_x and k_y non-zero will end up on one of the corners.

R-2.5

A projected band structure in a strip-geometry (PBC along x and OBC along y) will be helpful for understanding the results, especially for momentum resolved results in Figs. 4 and 5.

A-2.5

We thank the Referee for this suggestion. We now added the spectra plot that will be helpful to understand the momentum-resolved NHSE in the supplementary material.

R-2.6

The NHS modes are usually associated with some topological invariants, such as the winding number. What is the topological invariant associated in this case?

A-2.6

This is a good question. The 1D NHSE has its topological origin in the 1D point-gap winding number of the energy spectrum in the complex energy plane. From one perspective, the momentum-resolved skin effect of our system in a cylinder geometry can be thought of as quasi-1D skin effect. In this interpretation the edge skin effects are totally determined by the *momentum-resolved* (quasi-)1D point-gap winding number. However, there is currently no well-accepted topological invariant like the 1D point-gap winding number that captures the full rank-2 NHSE in 2D. So for our system under a square geometry, we can only say that rank-2 skin effect is closely related to the momentum-resolved 1D point-gap winding number along both the x and y directions. Finding a general classification of 2D NHSEs of various types using topological invariants is an interesting open question, but one that is beyond the scope of our article.

To clarify these points we added a section in the supplementary material to show the energy spectrum and discuss the topology of the rank-2 NHSE in a quasi-1D geometry.

Once the above points are addressed, I believe the manuscript can be accepted for publication.

We hope that we have addressed all of the referees questions appropriately.

REVIEWERS' COMMENTS

Reviewer #1 (Remarks to the Author):

Through this round of resubmission, the authors have tried to address the criticism that has come up in the first round of reports. My strongest concern had been the discrimination between back then Ref. 18 Hofmann et al., which is now Ref. 20 in the revised manuscript, and the proposal communicated in this manuscript.

The authors, I believe, do a good job in the response to the referees in highlighting the differences and similarities. It is still quite clear that both proposal are very similar - a reciprocal skin effect in two spatial dimensions. Both proposals have advantages and disadvantages. The biggest plus for the reciprocal skin effect in this manuscript is the resolution of 2d momenta instead of 1d momenta, and the existence of non-trivial corner modes. The biggest plus for Ref. 20 is the presence of exceptional points, which makes Ref. 20 more interesting and potentially more relevant for applications related to the anomalous spectral sensitivity around exceptional points.

In the manuscript, the comparison to Ref. 20 still is quite hidden. There is just a footnote on page 10 and one sentence on page 5. No effort is taken in highlighting the similarities with Ref. 20. Ultimately, however, it is the authors' decision how they will contextualize their work with preexisting works, so I do not want to impose myself on that any further.

I believe the work is well executed and innovative, and that the authors are excellent renowned contributors to the field. I do believe, however, that in terms of novelty and possible applications such as momentum-mode selective skin effect phenomena and skin effect despite reciprocity, a decent fraction of novelty contained in this manuscript is clearly covered by Ref. 20. Whether the degree of novelty and originality is still sufficient to warrant publication in Nature Communication I take a rather neutral stand. Given, however, how successful Ref. 20 apparently has been from its citation metrics, I would acknowledge the relevance of this kind of skin effect realizations, and lean towards publication in Nature Communications.

Reviewer #2 (Remarks to the Author):

The authors have answered my questions satisfactorily. I recommend the publication of this work.